# The Experience of Patient Safety Error for Nursing Students in COVID-19: Focusing on King’s Conceptual System Theory

**DOI:** 10.3390/ijerph20032741

**Published:** 2023-02-03

**Authors:** Mi Ok Song, Suhyun Kim

**Affiliations:** Department of Nursing, Nambu University, Gwangju 62271, Republic of Korea

**Keywords:** patient safety, risk management, COVID-19, nursing education, content analysis

## Abstract

Some nursing students experience errors related to patient safety, such as falls, medication administration errors, and patient identification errors during clinical practice. However, only a few nursing students report errors during clinical practice. Accordingly, the present study aimed to investigate patient safety errors that nursing students experience during clinical practice in the context of the COVID-19 pandemic. This study conducted in-depth interviews with 14 candidates for graduation from the Department of Nursing at a university in South Korea. In addition, after transcribing the collected data, a directed content analysis for the data based on King’s interacting system theory was performed. As a result, four core categories were identified: (i) nursing students’ perception of patient safety error occurrence, (ii) interaction between nursing students and others, (iii) interaction between nursing students and organizations, and (iv) nursing students’ training needs related to patient safety errors. Consequently, this study identified the patient safety error-related experiences of nursing students during clinical practice during the COVID-19 pandemic. The results suggest that in the future, nursing education institutions must establish a system for nursing students to report patient safety errors during clinical practice for patient safety education and develop practical and targeted education strategies in cooperation with practice training hospitals.

## 1. Introduction

Patient safety accounts are critical in healthcare settings, and medical errors by healthcare workers put patients’ safety at risk. Medical errors were reported to be the third main cause of death in the United States, after cardiovascular problems and cancer [1]. Patients have a 1 in 300 chance of being harmed while receiving medical care. Notably, 42.7 million adverse events occur worldwide, and patient harm accounts for the 14th major cause of morbidity and mortality, making patient safety a major global public health issue [2]. In addition, the coronavirus disease (COVID-19) pandemic has caused a global social crisis and directly affected healthcare systems [3]. Since the COVID-19 pandemic, nurses have been assigned to various departments for nursing COVID-19 patients. However, due to their excessive workload, the quality of nursing care is deteriorating, and patient safety-threatening errors are occurring increasingly [4].

More than 50% of healthcare workers commit errors that affect patients during their clinical career [5]. In particular, 49–53% of newly graduated registered nurses (RNs) with less than a year of experience are involved in medical errors [6]. It was found that during clinical practice, most nursing students experience hazardous situations that threaten patient safety during clinical practice, with 28–30% of nursing students experiencing medication administration errors [7]. In addition, 39% of nursing students made errors with no harm to patients during clinical practice [8], while 29% witnessed adverse events, wherein 85% of those errors were harmful to patients [9]. Moreover, their experiences observing patient safety accidents during clinical practice negatively affect the reporting intention of patient safety accidents [10].

Error reporting is one of the most crucial factors for promoting patient safety and healthcare quality [11]. However, the contents related to error reporting are rarely addressed in nursing education [12] though the PSE experience of nursing students in the clinical field may critically impact the transition of new nurses to professional nurses [13]. Nevertheless, most studies have been conducted on reporting medication administration errors by healthcare workers [14]; however, there is a need to conduct studies on error reporting committed by nursing students. In particular, nursing students must receive education on patient safety prior to clinical practice in South Korea [15]. However, it was found that 44% of nursing students have experienced PSEs during clinical practice, but most did not report them [16,17]. In this regard, it is necessary to seek strategies for effective patient safety education by investigating the experiences of patient safety errors (PSE) that nursing students encounter during clinical practice [18].

Most studies on PSEs have been conducted with nurses [19,20,21]. However, the error experience of nursing students, who are still training to become nurses, is different from that of nurses [22]. Furthermore, nursing students who are about to graduate may have a different clinical practice experience than before due to COVID-19 [23,24]. Accordingly, considering the situational context of COVID-19, this study aimed to explore and understand nursing students’ PSE-related experiences to prepare primary data for developing an effective patient safety education program for nursing students.

## 2. Materials and Methods

### 2.1. Study Design

This study was a qualitative study in which Hsieh and Shannon’s [25] directed content analysis approach was applied to understand and explore the PSE experiences of nursing students. The directed content analysis approach is a theoretical framework but extends the theory conceptually. In addition, this approach uses core concepts or variables derived from existing theories or studies as categories of initial coding [25].

### 2.2. Theoretical Framework

This study aims to understand and analyze the PSE experiences of nursing students based on King’s theory [26]. This theory includes the interacting dimensions of humans as personal, interpersonal, and social systems. Therefore, in this study, major categories were composed of personal PSE experiences of nursing students, experiences of interactions related to PSE during clinical practice, and experiences with hospitals (clinical practice institutions) and universities (nursing education institutions). In the growth and development section, goal attainment and outcomes by the transaction process mean an effective patient safety education program to be developed in the future; thus, education demands were included as a major category.

### 2.3. Participants

This study was conducted with 14 candidates for graduation from the Department of Nursing attending a university located in South Korea. The study participants were 13 males and 1 female, aged 24–26 years old. The inclusion criteria [27] were those who had experienced committing or witnessing PSEs and had completed all 1000 h of clinical practice because, in Korea, nursing students must complete a minimum of 1000 clinical practice hours to graduate [15]. Accordingly, all the participants received patient safety education obligatorily before clinical practice and were nursing students who had experienced more than 1000 h of clinical practice.

### 2.4. Recruitment

The study participants included those who met the inclusion criteria through a recruitment announcement in the university and were chosen using the purposive sampling method until a saturation state was reached in which no new categories were drawn during interviews. Data were collected using unstructured individual in-depth interviews for a month in April 2021. Interviews were conducted at the time and place desired by the participants, and face-to-face interviews were conducted with five participants. In comparison, Zoom interviews were conducted with nine participants. Face-to-face interviews were conducted once per person, each lasting about 40–80 min, and the researchers and participants wore a KF94 mask due to COVID-19. During the interview, the contents were recorded with the participant’s consent, and within 24 h after the interview, the contents were transcribed by experienced assistant researchers. The main and supporting interview questions are summarized in Table 1.

### 2.5. Data Analysis

In this study, King’s [26] conceptual system for nursing was used as a conceptual framework, and data were analyzed with directed content analysis [25]. First, the coding category in the key concepts from King’s theory was identified: personal systems, growth and development, interpersonal systems, social systems, and interactions. Perception, an experience in the personal system, was defined as “nursing students’ perception of PSE occurrence”. Interpersonal systems and interactions were defined as “interaction between nursing students and others”, and “others” included colleagues, nurses, and clinical practice advisors. Social systems and interactions were defined as “interaction between nursing students and organizations”, and “organizations” included a hospital, a clinical practice institution, and a university, an educational institution. In addition, growth and development were defined as “PSE education to be developed later” and included content corresponding to PSE education demands.

Second, the transcribed data were read independently to comprehensively understand the nursing students’ PSE-related clinical practice experiences and their PSE educational needs. Any text related to the four core categories was highlighted, coded, and assigned to the corresponding categories. The four core categories were derived from King’s theory and included nursing students’ perception of PSE occurrence, the interaction between nursing students and others, the interaction between nursing students and organizations, and nursing students’ training needs related to PSE. After coding, the logical connection and hierarchy between the categories and subcategories, such as whether the subcategories corresponding to each category were connected correctly, were reviewed, and whether the subcategories were composed of appropriate codes. In this way, a final analysis of the interviews was conducted separately, and then an agreement was reached after comparing and analyzing the results.

### 2.6. Researcher Preparation

The researchers have more than 10 years of experience as clinical nurses and have taught nursing theory and practice to nursing students as professors at universities for 8 years, allowing them to have the sensitivity to understand the PSE experiences of the study participants fully. In addition, the researchers have regularly participated in qualitative research seminars, developed their capacity for conducting qualitative research, and conducted several qualitative studies.

### 2.7. Data Analysis Study Validity and Rigor

Considering the rigorous evaluation standard of qualitative research, efforts were made to secure the trustworthiness of Lincoln and Guba [28] when analyzing the data. The interview contents of the participants were repeatedly checked to determine if they were consistent with the transcribed materials while continuously comparing them to ensure the truth value. Meanwhile, to ensure applicability, the study results were shown to three nursing students who did not participate in this study and checked for whether the results were similar to their own experiences and if they were meaningful. The data analysis process and procedures were strictly complied with, and this study’s results were shown to three qualitative research experts for research process evaluation to ensure consistency. Lastly, the results excluded preunderstanding and prejudice as much as possible, and analytical memos were recorded after the interview to maintain neutrality.

### 2.8. Ethical Considerations

This study was approved by the Institutional Review Board (IRB) of the primary investigator’s university (IRB No. 1041478-2020-HR-002). Upon recruiting the study participants, those who voluntarily expressed their intention to participate in this study were selected as participants. This study’s purpose and participation guidelines were explained to the participants, and written consent for study participation was obtained afterward. In addition, it was explained to the participants that the contents would be recorded during the interview and that they could stop the interview at any time if they wished. After the interview, all the participants were given return gifts as a reward for participating in this study.

## 3. Results

As a result of the study, four categories were derived that present major perspectives on the nursing students’ experience of PSE in clinical practice: (i) nursing students’ perception of PSE occurrence, (ii) interaction between nursing students and others, (iii) interaction between nursing students and organizations, and (iv) nursing students’ training needs related to PSE. The findings of this research are presented in Table 2. Four categories, twelve subcategories, and thirty-four codes were identified.

### 3.1. Nursing Students’ Perception of PSE Occurrence

#### 3.1.1. Fear

This subcategory was composed of three codes: “Worrying about patient’s condition”, “Afraid of being blamed”, and “Afraid of being responsible”. The participants were initially concerned about patients’ conditions but feared being blamed or taking responsibility for the situation.

“*I was worried about the patient’s condition once the accident occurred. I was afraid that the patient would be wrong because of me.*”(Participant 3)

“*I am afraid of being scolded when problems come up to patients because of my fault.*”(Participant 1)

“*I was afraid of being responsible for the accident after it occurred. What can I do for that?*”(Participant 5)

#### 3.1.2. Guilt

This subcategory was composed of two codes: “Feeling like it’s my fault for not checking accurately” and “Feeling like it’s my fault for not paying careful attention.” When PSEs occurred, the participants recognized that the cause of the errors lay within themselves. In other words, they believed that PSE situations occurred because they did not accurately check the drugs or patients or because of their negligence.

“*It’s my fault for not checking the patient’s registration number correctly and giving the drug to another patient.*”(Participant 7)

“*The patient fell while I was careless for a moment, which was because I didn’t watch the patient carefully.*”(Participant 3)

#### 3.1.3. Psychological Conflict

This subcategory was composed of two codes: “Obviously thinking I should report accidents, but don’t report them” and “Determining whether or not to report accidents depending on the patient’s condition”. The participants were taught to report PSEs when they occur in the orientation held before the start of clinical practice, but they did not do it and suffered psychological conflict.

“*When an accident comes up, I was taught to report it obviously. But I did not report it in real situations.*”(Participant 14)

“*I considered if I should report the accident. But, after watching the situation first, I did not report it unless the patient’s life was in danger.*”(Participant 8)

#### 3.1.4. Raising Awareness of the Risk of Patient Infection

This subcategory comprised two codes: “Washing hands more thoroughly than before” and “Listening to information related to patient infection”. The participants became especially sensitive to infection due to PSEs during clinical practice in the context of the COVID-19 pandemic. Accordingly, the participants washed their hands thoroughly, making efforts to prevent patient infection themselves and to carefully listen to information related to patient infection during handover time.

“*I’ve definitely been washing my hands thoroughly since the outbreak of COVID-19.*”(Participant 12)

“*During handover time, I listen carefully to patient infection information that I was not interested in before.*”(Participant 10)

### 3.2. Interaction between Nursing Students and Others

This category was composed of two subcategories: “Considering whether or not to report accidents depending on interests with the person involved in the accidents” and “Difficulties in communication to nurses”. The participants first decided whether or not to report PSE situations based on their existing interpersonal relationships by considering the interaction with the person involved in the accidents. In addition, they found it difficult to communicate PSE situations with nurses when detecting errors.

#### 3.2.1. Considering Whether or Not to Report Accidents Depending on the Interests of the Person Involved in the Accidents

This subcategory was composed of three codes: “Not reporting patient safety accidents if a close friend causes them”, “Reporting patient safety accidents if a friend from another school causes them”, and “Cannot be said if nurses cause patient safety accidents”. When colleagues caused patient safety accidents, the participants considered whether or not to report the accidents according to their closeness level with their colleagues. When nurses caused patient safety accidents, the nursing students took an attitude of acquiescence even though they recognized the situation.

“*If I report mistakes of my close friend to nurses, the friend gets embarrassed. So I didn’t report it.*”(Participant 5)

“*In principle, it is correct to report PSEs. If someone I don’t know well makes a mistake, I quietly report it to nurses. A nursing student from another school gets the wrong drug in a nebulizer, then immediately I report it to a nurse.*”(Participant 2)

“*How can I, a nursing student, report the mistakes of nurses? If it doesn’t harm a patient’s life, I just ignore it.*”(Participant 11)

#### 3.2.2. Difficulties in Communicating with Nurses

This subcategory was composed of three codes: “Missing opportunities to communicate with nurses due to their busy schedule”, “Having no idea how to talk about the mistakes of nurses”, and “Hardly meeting preceptor nurses during clinical practice”. Nursing students could not tell nurses about PSEs even when patient safety was threatened as nurses’ work was increased and nursing personnel was short due to COVID-19. In addition, nursing students felt embarrassed to ask questions about nurses’ behaviors. Moreover, without proper guidance and supervision from preceptors who guide clinical practice, nursing students did not have an opportunity to ask questions about PSEs or discuss them.

“*I want to ask nurses if it is fine to do this for a patient, but they are all too busy to ask it.*”(Participant 4)

“*If I tell a nurse that injecting without cleaning three-way valves would infect patients, they will consider me as a strange student. How can I point out their fault?*”(Participant 13)

“*I didn’t even have a chance to ask preceptors who guide clinical practice. I hardly met them, so I couldn’t ask because their work schedules didn’t fit our practice schedules.*”(Participant 2)

### 3.3. Interaction between Nursing Students and Organizations

This category was composed of three subcategories: “Culture of clinical practice institutions that make it difficult to report accidents easily”, “Reinforced demands of clinical practice institutions for preventing infection due to COVID-19”, and “Ineffective patient safety management education in universities”. The participants could not even consider reporting on PSEs in the atmosphere of reproach or rebuke in the clinical practice field, and they had to comply with stronger infection prevention regulations than before due to COVID-19. In addition, they received education on patient safety management at the university before clinical practice but faced the reality that the education could not be applied to the clinical field.

#### 3.3.1. Culture of Clinical Practice Institutions That Makes It Difficult to Report Accidents Easily

This subcategory comprised two codes: “Atmosphere that blames nurses” and “Atmosphere that stigmatizes nurses as criminals”. The nursing students came into contact with the organizational culture of hospitals by directly or indirectly experiencing PSE situations during clinical practice. In this culture, the nurses who created PSE situations had to take responsibility for all the situations. Thus, the nursing students experienced the nurses being blamed and stigmatized as criminals when the patient’s condition deteriorated.

“*In this hospital, when something goes wrong with a patient, the nurse in charge is scolded.*”(Participant 3)

“*A patient had a fall, and a senior nurse was very angry with the nurse in charge.*”(Participant 9)

#### 3.3.2. Reinforced Demands of Clinical Practice Institutions for Preventing Infection due to COVID-19

This subcategory was composed of three codes: “Emphasizing thorough mask wearing and thorough hand washing”, “Patient contact limited, including vital signs”, and “Repetitive coronavirus testing conducted before and after clinical practice”. The nursing students frequently received preventive regulations from clinical practice institutions due to COVID-19. Clinical practice institutions asked the nursing students to replace their KF94 masks with new ones after they arrived in the institutions, thoroughly monitored their hand washing, and restricted all their contact with patients. In addition, the institutions asked the nursing students to submit their COVID-19 polymerase chain reaction (PCR) test results to the relevant institution before and after the start of clinical practice.

“*Previously, nurses did not thoroughly conduct six steps for hand washing, but after the outbreak of COVID-19, they are all washing their hands thoroughly. Wearing a mask is also a must.*”(Participant 8)

“*I went to the hospital for clinical practice, but I couldn’t do anything because I couldn’t touch patients for their safety.*”(Participant 2)

“*For clinical practice, we are undergoing coronavirus testing every week.*”(Participant 4)

#### 3.3.3. Ineffective Patient Safety Management Education in Universities

This subcategory was composed of four codes: “Receiving compulsory education perfunctorily on patient safety before clinical practice”, “Education on patient safety accidents received, but hard to apply it to practice”, “Group education received using Zoom, but not understanding it”, and “Education on the response system for coronavirus infection changed frequently and thus delivered through social networking services (SNS) used by Koreans like Kakao Talk”. The nursing students received education on patient safety obligatorily before clinical practice, but because the same education was repeated, they perfunctorily participated in it. In addition, in PSE situations, they could not determine how to cope with them in practice. In particular, due to COVID-19, they received patient safety education in non-face-to-face groups, so they could not concentrate on the lecture and understand all the contents. Patient safety education covers various topics, from patient safety accidents to falls and medication administration, but the part on patient infection was especially emphasized due to COVID-19. The nursing students were not provided with a manual on how to deal with infection; thus, practical guidance professors frequently delivered the changes to them using SNS.

“*I received education on patient safety accidents at the orientation before clinical practice, but I don’t understand it well.*”(Participant 6)

“*I’ve heard of adverse events and sentinel events, but when a patient safety accident occurred, I couldn’t remember what they were.*”(Participant 4)

“*I don’t know how the patient safety reporting system is organized.*”(Participant 9)

“*I received patient safety education, but I don’t know what to do first when a patient falls at the clinical field, or when I find out too late that a wrong medicine has been delivered to a patient.*”(Participant 12)

### 3.4. Nursing Students’ Training Needs Related to PSE

This category was composed of three subcategories: the training content, the teaching method, and the learning method. The participants presented detailed opinions to develop systematic and effective PSE education based on their experiences.

#### 3.4.1. Training Content

This subcategory was composed of three codes: “Patient safety report system according to clinical practice institutions”, “How to deal with patient safety accidents by the situation (medication administration accident, fall, preparation of nebulizer drugs, etc.)”, and “Self-assertive communication skills”. Whenever clinical practice institutions change, their hospital systems differ slightly. Accordingly, the participants wanted to learn in detail about patient safety reporting systems by the institution. In addition, they wanted to know how to respond to each situation to immediately cope with situations such as medication administration accidents and falls that they experienced. Notably, they experienced difficulties communicating with nurses and wanted to learn self-assertive communication skills to convey their thoughts and opinions accurately.

“*Clinical practice institutions keep changing, and we want to know exactly how the PSE reporting system is organized for each hospital.*”(Participant 1)

“*In situations such as when a wrong medicine is given, when a fall occurs, and when a wrong medicine goes into a nebulizer, what should we do first, and what is the next nursing performance? I hope these contents are manualized.*”(Participant 7)

“*I want to know communication methods to accurately conveys what I want to say. Then, when a PSE situation occurs after I become a new nurse, I think I will be able to say accurately without trying to hide the situation.*”(Participant 2)

#### 3.4.2. Teaching and Learning Method

This subcategory was composed of seven codes: “Lecture-based approach to convey basic concepts”, “Sharing detailed cases (case study)”, “Simulation learning that mimics actual clinical settings”, “A form that allows repetitive learning”, “A form that allows non-face-to-face access”, “E-learning contents accessible by the situation”, and “Contents considering applicability to practice”. The basic concepts of PSE can be learned in a traditional lecture format, but the participants wanted to know complex clinical cases to remember them well. In addition, they wanted to do better nursing performance in clinical practice by directly experiencing PSE situations similar to actual ones. The participants wanted an educational program that allows for repetitive learning at the time and place they want. Moreover, they proposed education programs in the form of E-learning education that shows how to cope with each situation and in a form allowing for an immediate application to practice after learning.

“*I think it would be good to clarify the concepts, such as sentinel events and adverse events, through lectures like now.*”(Participant 4)

“*I think it would be helpful for education if there is a description of the situation in detail rather than a written phrase ‘when a fall occurs.’ For example, I want to see a specific explanation or situation, such as ‘a patient tried to get up after surgery and fell down.’*”(Participant 9)

“*If I have the experience of coping with a virtual situation in which PSEs occur before clinical practice, I think I would be reminded of how to act when faced with PSE situations in clinical practice.*”(Participant 2)

“*I can’t remember the education that I received once in groups. I want to be able to have a repetitive access to the education to learn it again when needed.*”(Participant 1)

“*I don’t think the COVID-19 situation is going to end easily. I hope that a non-face-to-face education program opened.*”(Participant 3)

“*I hope that educational content by situation will be produced in the form of E-learning that can be accessed and learned whenever I want to know how to deal with PSE situations I experienced.*”(Participant 1)

“*I think it would be nice if an education program is produced in a form that we can really apply to practice after education, rather than a perfunctory group education.*”(Participant 5)

## 4. Discussion

The present study was conducted to prepare basic data for developing an effective patient safety education program for nursing students by investigating and understanding their PSE experiences in consideration of the situational context of COVID-19. The nursing students were alerted to the risk of patient infection in the context of the COVID-19 pandemic and also felt that error reporting was difficult due to fear of error reporting, communication difficulties, and the culture of clinical practice institutions. In addition, they presented the need for error-reporting education that utilizes appropriate teaching–learning strategies and can be applied in practice.

First, looking into the nursing students’ experiences that correspond to the personal system of King’s theory, upon a patient safety accident, they experienced fear of being blamed and guilt for their carelessness, even though they were concerned about the patient’s condition. In addition, they experienced psychological conflict over whether or not to report errors depending on the patient’s condition. It was found that nursing students were alerted to the risk of patient infection due to the COVID-19 pandemic and made efforts to prevent it, such as hand washing. Previous studies have shown that nursing students feared being blamed when experiencing errors during clinical practice [8,16] and thought the errors affected their trust in themselves. The negative experiences with PSEs stem from a personal approach, which assumes that the errors arise from individual unsafe actions such as inattention, forgetfulness, and negligence. Therefore, PSE experiences must be approached systematically, focusing on the causes and solutions of errors [29]. Moreover, such psychological stress may have been aggravated due to frequent policy and guidelines changes due to the COVID-19 pandemic [24]. These negative experiences of nursing students can be an obstacle to error reporting. Thus, to prevent this, continuous and in-depth patient safety education, a patient safety culture, and a safe learning culture that encourages questions based on teamwork and trust will be required. In addition, error reporting needs to be promoted through the use of role models related to patient safety management or the development of strategies to reduce negative emotions about error experiences by allowing nursing students to learn emotional preparedness related to error experiences through debriefing about errors and acknowledging their emotions.

Second, looking into the interactions of nursing students in the interpersonal relationship system, it was found that they decided whether to report errors according to their interest in others. They needed help communicating errors with nurses due to their workload. Previous studies have revealed that when nurses commit an error, nursing students differentiate the type of error report depending on the situation and make formal or informal reports [21]. When medical students experience or find an error, they also decide whether to report it according to its subject, the witness’s existence, and the result of the error, and they are affected by their emotions at that moment [30], which is consistent with this study’s results. These results support the importance of an educational approach that can make nursing students properly deal with errors when they commit or discover them. Therefore, it is necessary to promote PSE learning by establishing a system where nursing students can report PSEs during clinical practice, report errors officially, and share them. Notably, nursing students have reported that they did not receive sufficient support and supervision from nurses and found it difficult to ask questions to them during clinical practice in the context of the COVID-19 pandemic [27]. This situation has made it difficult for nursing students to report errors even if they find errors during clinical practice, and it also has made them experience negative emotions, such as stress. It is believed that this experience has negatively impacted future error reporting. Moreover, a lack of nontechnical skills causes 70–80% of PSEs [31], and nontechnical skills, such as situational awareness and communication, are important to improve patient safety [32], and education and training are necessary to communicate accurately even in these situations. In addition, the accurate recognition of patient safety is achieved by detecting errors and collecting necessary information, which can be directly related to communication for patient safety. Confidence and opinion about patient safety are not established in a short time [33]; therefore, repeated education is required to train accurate situational recognition and communication.

Third, as a result of exploring the interaction of nursing students in the social system, it was found that nursing students realized a culture that makes reporting PSEs when they occur difficult because of the reinforced policy of clinical practice institutions for infection control due to COVID-19, and they were not familiar with PSE reporting because of ineffective patient safety management education in universities. Error reporting significantly correlates with patient safety culture [34]. However, clinical nursing teachers and supervisors in clinical practice institutions sometimes show an indifferent attitude towards minor nursing safety events, regard these situations as personal problems, and conceal or do not report them [35]. In addition, it was found that when errors occurred, nurses were hesitant to report them because they experienced blame and stigma and did not receive any support from the organization, resulting in negative results [19]. This indicates that a patient safety culture that supports error reporting must be established. Moreover, error experience affects the experience of a nurse even a long time after the error occurs [21], so it is believed that the error experience of nursing students can affect their future careers as a nurse and their patient safety management activities. Therefore, clinical practice institutions and nursing education institutions must make efforts so that nursing students can discover, report, and properly deal with errors. The participants in this study realized the strengthened policies of clinical practice institutions due to COVID-19, and it turns out that this is because healthcare institutions emphasized controls and procedures for safety management to prevent errors in the context of the COVID-19 pandemic [36]. These measures support the results of this study. This situation is believed to have served as an opportunity to reinforce nursing students’ motivation to follow patient safety procedures and regulations. However, the participants reported that they did not know well about PSE reporting, so it is necessary to strengthen prior education on the concept, purpose, scope, and countermeasures of patient safety before clinical practice and theoretical classes. In addition, through case studies on previously reported PSE cases, learning about cause analysis, the derivation of improvement plans, and error reporting should be made within the nursing curriculum.

Fourth, for PSE education, the nursing students wanted education on practical methods to deal with errors, experiential learning through detailed clinical cases, and education that allows for repeated learning regardless of time and place. Notably, it was found that nursing students need to gain knowledge about error reporting and acquire patient safety-related knowledge mostly in pre-practice education conducted by clinical practice institutions. However, because error management and education levels vary among institutions, patient safety education for nursing students must be conducted before clinical practice [35]. Error reporting education using lectures, video case-based studies, and simulations is being conducted in medical education [37]. However, there is a lack of practical simulation training opportunities in the nursing curriculum to deal with errors, near-misses, and adverse events, leading to a lack of nursing performance capacity to identify errors, report them appropriately, and address them within clinical practice [38]. Therefore, nursing education institutions must prepare detailed and consistent reporting guidelines and collect data accurately and continuously to use as learning materials. In addition, studies are being conducted to investigate changes in the knowledge and awareness levels of patient safety before and after patient safety management education. However, they are temporary studies, and it needs to be confirmed whether this knowledge is retained in nursing students [38]. Moreover, the criteria for the period, contents, qualities of instructors, and teaching and methods for patient safety education vary from study to study, so it is difficult to determine which method is the best [8,38,39]. Patient safety education should include education to develop not only clinical competency but also sociocultural competency. However, the nursing students had low confidence in their sociocultural competency compared to their clinical competency, and patient safety-related contents need to be sufficiently addressed within their learning process [40]. Therefore, patient safety education for nursing students should be conducted as a stand-alone subject focusing on error reporting, not a fragmented matter within various programs. It is necessary to seek ways to promote learning within a situational context and enable self-learning based on understanding the cause of errors.

It is necessary to be careful about the interpretation because these results are based on data drawn from 14 participants considering Korea’s contextual situations and cultural characteristics. In addition, the interview of this study was conducted using an online digital platform due to the COVID-19 situation, which may have affected the interactions between the participants and researchers. However, this online digital platform had the advantage of enabling visual feedback [41,42] and immediately checking and analyzing the interview files. The results of this study can help explain the error experiences of nursing students due to the COVID-19 pandemic and its impact on learning.

## 5. Conclusions

This study was conducted to lay a groundwork for understanding the PSE experiences of nursing students and developing a patient safety education program in the context of the COVID-19 pandemic. The results suggest that an effective education program in nursing education institutions is needed for appropriate coping with and reporting of PSEs when nursing students experience them. For nursing students’ patient safety education, educational institutions need to prepare strategies to improve the perception of errors and negative experiences by establishing a patient safety culture. In addition, patient safety education should be conducted continuously, even in the next pandemic after COVID-19, through the accurate recognition and communication of error situations, error reporting learning based on actual cases of error reporting, error reporting education using various teaching and learning methods, and the development of educational contents that are unaffected by time and place. There is a need for healthcare facilities to conduct continuous self-assessments to improve patient safety and also minimize errors. In addition, providing further support to staff who have made errors is needed to determine if they need any psychological support and to further understand the cause of the errors. Moreover, similar studies to increase the data within the study setting and to develop strategies to help improve the situation are also needed.

## Figures and Tables

**Table 1 ijerph-20-02741-t001:** Examples of questions related to patient safety error (PSE) experiences used during the interview sessions with the participants.

Category	Items
Main question	Tell me about your PSE-related experiences during clinical practice.
Supporting questions	“What did you feel in that situation?” “What was your thought in that situation?” “What interactions did you have with people around you in that situation?” “What interactions did you have with hospitals or educational institutions in that situation?” “What educational content should be included for better PSE education?” “What kind of approach should be taken for effective PSE education?”

**Table 2 ijerph-20-02741-t002:** Categories and subcategories about nursing students’ experience with patient safety errors.

Category	Subcategory
Nursing students’ perception of PSE occurrence	Fear
	Guilt
	Psychological conflict
	Raising awareness of the risk of patient infection
Interaction between nursing students and others (colleagues, nurses, professors, etc.)	Considering whether to report accidents depending on interests
	Difficulty communicating with nurses
Interaction between nursing students and organizations (hospitals, universities)	Culture of clinical practice institutions make it difficult to report accidents easily
	Reinforced demands of clinical practice institutions for preventing infection due to COVID-19
	Ineffective patient safety management education in universities
Nursing students’ training needs related to PSE	Training content
	Teaching and learning method

## Data Availability

The data presented in this study are not publicly available due to privacy reasons.

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
