# Peer review of "The Experience of Patient Safety Error for Nursing Students in COVID-19: Focusing on King’s Conceptual System Theory"

_ijerph, 2023, doi:10.3390/ijerph20032741_

Round 1

Reviewer 1 Report

Introduction line 26-27: Since the COVID-19 pandemic, nurses have been assigned to various departments for nursing COVID-19 patients. line 28-29 patient safety–threatening errors are occurring increasingly; revise this statement. line 30 critical in health-care setting. line 33 and 34 write 1 in word one (1). line 50 promoting patient safety. line 56, however there is a need to conduct studies on error reporting committed by nursing students. line 57 nursing students must receive education on patient safety prior to clinical practice in South Korea. line 63 who are still training to become nurses. line 64 who are about to graduate (delete recently).

study design: line 72, analysis approach was applied. 

participants: line 89 comprised of 13 males and one (1) female, aged 24–26 years old. 

recruitment: line 100 and face-to-face interviews conducted with five participants. line 101 In comparison, zoom interviews were conducted with nine participants. 

conclusion: 504-506 please revise the statement, it is not so clear. 506-508 there is a need for healthcare facilities to conduct continuous self-assessment to improve patient safety and also minimize errors. there is also a need to further support staff who have made errors to see if they need any psychological support, and also further understand the cause of the errors. there is also a need for simmilar studies to increase data within the study setting and also a study that will develop strategies that can help improve the situation.

Please add recommendations.

Please attach ethical approval and data collection tool if they were not included. 

Reviewer 2 Report

Overall, it is an interesting and well-written paper. Patient safety errors could occur at any time and not specifically during the pandemic. I agree that it could increase during the pandemic for various specific reasons, but based on the paper, shows that the experiences and reasons highlighted by the participants are mostly general. Therefore, some clarifications are helpful and the first paragraph in the introduction needs to be better linked to the rest of the content.

Also, it is not clear what kind of errors the participants made (severity of the errors) and when they occurred. Are the experiences explored based on retrospective experiences or just those that occurred during the pandemic (precisely in the time frame)?

Do the inclusion criteria apply to students with experience of committed or witnessed patient safety errors, or is it assumed that all students had such PSE experience?

Otherwise, it gives the impression that all students could have experienced errors since data saturation could be reached with only 14 participants.

 The data analysis is based on the pre-determined framework and the data has been fitted into the four core categories, so it was not identified purely on the basis of analysis, this needs fine-tuning in writing and avoiding repeating the same content at many sessions or paragraphs.

Other comments:

·       The title needs refinement to show if the content is related to patient safety errors during the COVID-19 pandemic.

·       Abstract -  do not include abbreviations

·       Check for typos (e.g. Page 2, line 50; Page 10, line 408)

·       Page 8, Line 312: SNS means what?

·       The limitation of generalisability is the nature of the qualitative study (Line 491). 
